# Morphology of foveal hypoplasia: Hyporeflective zones in the Henle fiber layer of eyes with high-grade foveal hypoplasia

Andreas Bringmann *, Thomas Barth, Focke Ziemssen

Department of Ophthalmology and Eye Hospital, University of Leipzig, Leipzig, Germany

* bria@medizin.uni-leipzig.de

## Abstract

### Background

Foveal hypoplasia is characterized by the persistence of inner retinal layers at the macular center. We evaluated using spectral-domain optical coherence tomography (SD-OCT) morphological parameters of the macular center of eyes with foveal hypoplasia and describe the presence of hyporeflective zones in the Henle fiber layer (HFL) of eyes with high-grade foveal hypoplasia.

### Methods

Eyes with foveal hypoplasia were classified into two groups: high-grade foveal hypoplasia with thick inner retinal layers at the macular center (thickness above 100 μm; 16 eyes of 9 subjects) and low-grade foveal hypoplasia with thinner inner retinal layers at the macular center (thickness below 100 μm; 25 eyes of 13 subjects). As comparison, SD-OCT images of normal control eyes (n = 75) were investigated.

### Results

Eyes with foveal hypoplasia displayed shorter central photoreceptor outer segments (POS), a thinner central myoid zone, and a thicker central HFL compared to control eyes. Eyes with high-grade foveal hypoplasia also displayed a thinner central outer nuclear layer (ONL) compared to eyes with low-grade foveal hypoplasia and control eyes. There was a negative correlation between the thicknesses of the central ONL and HFL in eyes with foveal hypoplasia; however, the total thickness of both ONL and HFL was similar in all eye populations investigated. Visual acuity of subjects with foveal hypoplasia was negatively correlated to the thickness of the central inner retinal layers and positively correlated to the length of central POS. In contrast to central POS, the length of paracentral POS (0.5 and 1.0 mm nasal from the macular center) was not different between the three eye populations investigated. The paracentral ONL was thickest in eyes with high-grade foveal hypoplasia and thinnest in control eyes. Hyporeflective zones in the HFL were observed on SD-OCT images of eyes with high-grade foveal hypoplasia, but not of eyes with low-grade foveal hypoplasia and control eyes. OCT angiography images recorded at the level of the HFL of eyes with high-grade

**Data Availability Statement:** All relevant data are within the paper and its Supporting Information files.

**Funding:** This research did not receive any specific grant from funding agencies in the public, commercial, or not-for-profit sectors.

**Competing interests:** The authors have declared that no competing interests exist.

**Abbreviations:** BCVA, best-corrected visual acuity; ELM, external limiting membrane; EZ, ellipsoid zone; GCL, ganglion cell layer; HFL, Henle fiber layer; IZ, interdigitation zone; MCC, Müller cell cone; OCTA, optical coherence tomographic angiography; ONL, outer nuclear layer; OPL, outer plexiform layer; PIS, photoreceptor inner segments; POS, photoreceptor outer segments; RPE, retinal pigment epithelium; SD-OCT, spectral-domain optical coherence tomography.

foveal hypoplasia showed concentric rings of different reflectivity around the macular center; such rings were not observed on images of eyes with low-grade foveal hypoplasia and control eyes.

## Conclusions

It is suggested that the hyporeflective zones in the HFL of eyes with high-grade foveal hypoplasia represent cystoid spaces which are surrounded by Henle fiber bundles. Cystoid spaces are likely formed because there are fewer Henle fibers and a thinner central ONL despite an unchanged thickness of both ONL and HFL. Cystoid spaces may cause the concentric rings of different reflectivity around the macular center in the HFL of eyes with high-grade foveal hypoplasia.

## Introduction

The fovea as the retinal region specialized for high-acuity color vision comprises the foveola ("little fovea") and the surrounding sloping foveal walls. The foveal region is characterized by the foveal pit, the presence of avascular and rod-free zones, a high density of cone photoreceptors, and elongation of the photoreceptor outer segments (POS) [1–4]. The fovea develops by a centrifugal displacement of the inner retinal layers, which results in the formation of the foveal pit (fovea interna), and a centripetal displacement of the outer retinal layers which causes a thickening of the central outer nuclear layer (ONL) and the formation of the fovea externa that is the cone-like arrangement of the thin elongated photoreceptor segments in the foveola [2, 4–6]. Horizontally or obliquely arranged Henle fibers compensate the spatial shift between the inner and outer retinal layers. The fovea externa is visible on histological sections and spectral-domain optical coherence tomography (SD-OCT) images at the inwardly inclined courses of the external limiting membrane (ELM) and ellipsoid zone (EZ) lines in the foveola.

The development of the foveal pit begins after fetal week 25 [6] and was proposed to proceed by three processes [4]: (i) Pit formation is initiated by a vertical contraction of central Müller cells. (ii) Thereafter, the pit is widened by the centrifugal displacement of ganglion cells produced by the tangential contraction of the astrocytic network in the nerve fiber layer and ganglion cell layer (GCL) around the pit; this results in a tilt of the inner retinal layers in the foveal walls and parafovea. (iii) The inner layers are erected by a centrifugal displacement of the tissue at the level of the outer plexiform layer (OPL) due to a horizontal contraction of Müller cell processes.

The foveal morphology in the normal human population shows substantial variations in respect to different morphological parameters like the peak photoreceptor density, the magnitude of the centrifugal displacement of inner retinal layers, the areas of the foveola and avascular zone, the depth of the foveal pit, the thickness of the foveola, and the steepness of the foveal walls [7–12]. The morphological variability of the fovea is considered to be a consequence of variations in the development phase [13]. Generally, a reduced magnitude of the centrifugal displacement of the inner layers results in smaller areas of the foveola and avascular zone, a lower depth of the foveal pit, increased thickness of the foveola, and decreased steepness of the foveal slope [4, 9].

A lack or reduction of the developmental displacements of the inner and outer retinal layers results in foveal hypoplasia. Foveal hypoplasia can be observed in different disorders like

oculocutaneous and ocular albinism, aniridia, achromatopsia, myopia, microphthalmos, and retinopathy of prematurity, and can occur isolated with no clear etiology [14–20]. The partial or full retention of inner retinal layers in the fovea of subjects with albinism is associated with increased foveal thickness, a shallow or absent foveal pit, and the absence of avascular and rod-free zones; foveal photoreceptors are large, loosely packed, and immature in shape [16, 21–23]. However, the morphology of the foveal pit and the photoreceptor density in the foveal center are highly variable in subjects with albinism; a nearly normal central photoreceptor packing and receptor elongation can be observed in some eyes with complete absence of a foveal pit [18, 24–28]. In the mean, increasing grades of foveal hypoplasia are correlated with worsening visual acuity [24, 26, 29, 30]; the average peak photoreceptor density is lower in subjects with albinism and foveal hypoplasia compared to normal control subjects [26, 27, 31].

It was described that the central Henle fiber layer (HFL) in eyes of subjects with albinism and foveal hypoplasia is thicker than in eyes of control subjects, likely because the central ONL is thinner [32]. Concentric macular rings of different reflectivity around the expected fovea can be observed at the level of the HFL on infrared reflectance, scanning laser polarimetry, en-face OCT, and ultra-wildfield fundus images of eyes with foveal hypoplasia [30, 32, 33]. The present study describes the presence of hyporeflective zones in the HFL on SD-OCT images of eyes with high-grade foveal hypoplasia which may represent cystoid spaces. We also describe that concentric rings of different reflectivity around the macular center can be observed on en-face OCT angiography (OCTA) images at the level of the HFL. We assume that the cystoid spaces are the cause of the concentric macular rings of different reflectivity in the HFL of eyes with high-grade foveal hypoplasia.

## Methods

This retrospective, single-center chart review followed the tenets of the 1964 Declaration of Helsinki and its later amendments, and was approved by the Ethics Committee of the Medical Faculty of the University of Leipzig (#143/20-ek). The ethics committee is registered as Institutional Review Board at the Office for Human Research Protections (registration number, IORG0001320/IRB00001750). We retrospectively reviewed charts of 89,317 subjects who were referred to the Department of Ophthalmology, University of Leipzig, Germany, between August 2008 and December 2019. We searched for charts which contained SD-OCT images of eyes with foveal hypoplasia indicated by the presence of inner retinal layers at the macular center. We found charts of 41 eyes of 22 subjects (11 females, 11 males; mean ± S.D. age, 31.3 ± 20.5 years, range, 6–68 years) which fulfilled this criterion. The cohort included 17 subjects with idiopathic foveal hypoplasia and three subjects with foveal hypoplasia secondary to oculocutaneous or ocular albinism (Figs 1Bd and 2C). Nystagmus was observed in two subjects. Subjects characteristics are presented in Table 1. The subjects gave their oral informed consent for their images and data to be analyzed and reported anonymously. SD-OCT images of 75 randomly selected normal control eyes of 75 subjects were also investigated (50 females, 25 males; mean age, 41.5 ± 19.8 years, range, 6–68 years); the selection criterium was the detectability of the central and nasal HFL and ONL on the images, i.e., the HFL displayed a higher reflectivity than the ONL. Medical chart review and fundus photographs examination was also done to exclude eyes with any retinal or choroidal pathology.

5.5-mm, 6-line radial SD-OCT scans of the macula were recorded with Heidelberg Spectralis OCT (Heidelberg Engineering, Heidelberg, Germany). OCTA images were recorded with Zeiss PLEX Elite 9000 (Carl Zeiss Meditec, Jena, Germany). Fundus images were recorded with the Nidek AFC-230 camera (Nidek, Aichi, Japan). Best-corrected visual acuity (BCVA) was determined using Snellen charts and is given in logMAR.

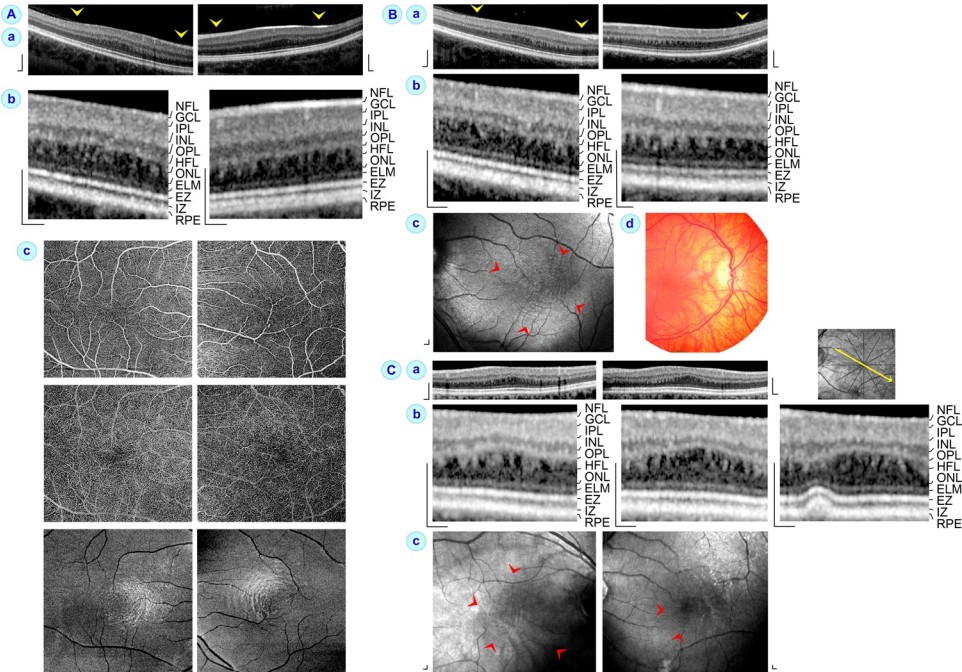

**Fig 1. High-grade foveal hypoplasia may be associated with hyporeflective zones in the central Henle fiber layer (HFL) and concentric rings around the macular center in en-face optical coherence tomographic angiography (OCTA) images at the level of the HFL.** The images show horizontal SD-OCT (**a**) and OCTA scans (**c**) through the macula of the right (*left side*) and left eyes (*right side*) of subjects 1 (**A**), 2 (**B**), and 3 (**C**). The images in **b** show SD-OCT scans of the central macula at higher magnification. **Ac.** OCTA images recorded at the superficial (*above*) and deep vascular plexus (*middle*), and at the level of the HFL (*below*). In **Bc** and **Cc**, the OCTA images were recorded at the level of the HFL. Note that the OCTA images recorded at the level of the HFL show shadows of overlying vessels. **Bd.** The fundus photograph of this eye of a subject with ocular albinism shows hypopigmentation of the RPE and visualization of large choroidal vessels. **Cb**, *right*: The druse in this eye with intermediate age-related macular degeneration caused an inward displacement of the retinal pigment epithelium (RPE), interdigitation zone (IZ), ellipsoid zone (EZ), external limiting membrane (ELM), and outer nuclear layer (ONL) which was associated with an absence of hyporeflective zones in the HFL at this site. *Yellow arrowheads* indicate adhesions of the partially detached posterior hyaloid at the macular tissue. *Red arrowheads* indicate concentric rings of different reflectivity around the macular center. Scale bars, 200 μm. GCL, ganglion cell layer; INL, inner nuclear layer; IPL, inner plexiform layer; NFL, nerve fiber layer; OPL, outer plexiform layer.

The thickness of individual retinal layers was manually measured using the Heidelberg Eye Explorer 2 software (Heidelberg Engineering) on horizontal SD-OCT images within ± 2.0 mm of the macular center (defined as the site of the lowest thickness of the inner retinal layers in eyes with foveal hypoplasia and the lowest thickness of the foveola of control eyes; Fig 3A). The following distances were measured: the distance between the retinal pigment epithelium (RPE)-Bruch's membrane interface and RPE-interdigitation zone (IZ) interface which roughly represents the thickness of the RPE; the distance between the RPE-IZ interface and the outer border of the ELM, representing the length of photoreceptors (receptor segments); the distance between the RPE-IZ interface and the outer border of the EZ, representing the length of POS; the distance between the outer border of the EZ and the outer border of the ELM, representing the length of photoreceptor inner segments (PIS); the thickness of the EZ of PIS; and the distance between the inner border of the EZ and the outer border of the ELM, representing the thickness of the myoid zone (MZ) of PIS. Furthermore, the vertical thicknesses of the ONL, HFL, and the different inner retinal layers, as well as the total retinal thickness (from the

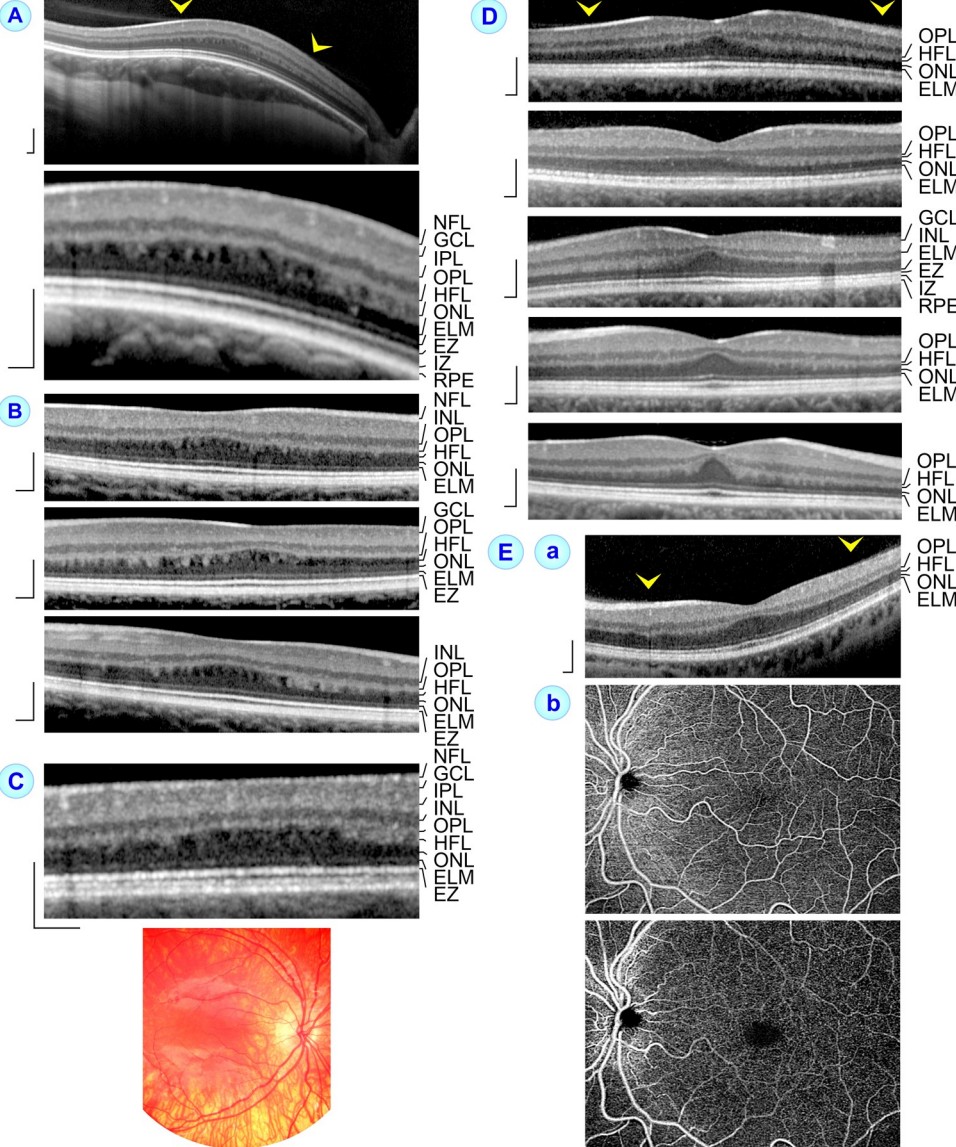

**Fig 2.** SD-OCT scans through the macula of eyes with high- (**A–C**) and low-grade foveal hypoplasia (**D, Ea**). **A, B.** Eyes with high-grade foveal hypoplasia and larger hyporeflective zones in the Henle fiber layer (HFL). **A.** The dome-shaped macula of the right eye of subject 4 likely resulted from scleral thickening in this myopic eye. The image *below* shows the central macula at higher magnification. **B.** Horizontal scan through the macula of the right eye (*above*) and horizontal (*middle*) and vertical (*below*) scans through the macula of the left eye of subject 5. **C.** SD-OCT scan through the macula of the right eye of subject 6 with oculocutaneous albinism, high-grade foveal hypoplasia, and smaller hyporeflective zones in the HFL. The fundus photograph *below* shows hypopigmented RPE with visualization of large choroidal vessels. **D, Ea.** Eyes with low-grade foveal hypoplasia and shallow foveal pits. The images show SD-OCT scans recorded in the right eyes of subjects 10 and 19, the left eyes of subjects 10 and 13, and the right eye of subject 21 (**D**), as well as the left eye of subject 14 (**Ea**). **Eb.** The optical coherence tomographic angiography images of the left eye of subject 14 were recorded at the levels of the superior (*above*) and deep vascular plexus (*below*). *Arrowheads* indicate adhesions of the partially detached posterior hyaloid at the macular tissue. Scale bars, 200 μm. ELM, external limiting membrane; EZ, ellipsoid zone; GCL, ganglion cell layer; INL, inner nuclear layer; IPL, inner plexiform layer; IZ, interdigitation zone; NFL, nerve fiber layer; ONL, outer nuclear layer; OPL, outer plexiform layer; RPE, retinal pigment epithelium.

Table 1. Characteristics of subjects with foveal hypoplasia.

| Subject No. | Sex/Age (years) | BCVA R/ L | Diagnosis |
|---|---|---|---|
| **High-grade foveal hypoplasia with larger hyporeflective zones in the HFL** | | | |
| 1 | F/17 | (0.1)/ (0.2) | R, L: optic nerve hypoplasia, nystagmus<br>L: cellophane maculopathy |
| 2 | F/34 | 0.1/0.1 | R, L: astigmatism, esotropia, ocular albinism |
| 3 | F/32 | 0.1/0.2 | R: dry age-related macular degeneration |
| 4 | F/68 | (0.7)/ (0.7) | R, L: oculocutaneous albinism, esotropia, myopia, amblyopia; L: no SD-OCT record |
| 5 | F/8 | n.d. | R, L: unclear subjective visual disturbance |
| **High-grade foveal hypoplasia with smaller hyporeflective zones in the HFL** | | | |
| 6 | F/9 | 0.4/1.3 | R, L: oculocutaneous albinism, esotropia |
| 7 | F/67 | (0.4)/ (0.1) | R: amblyopia, exophoria<br>R, L: cataracta senilis incipiens |
| 8 | M/16 | 0.5/0.5 | L: intermittent exotropia, astigmatism hyperopicus |
| 9 | M/65 | 0.0/(0.3) | L: epiretinal membrane-mediated thickening of the macular tissue |
| **Low-grade foveal hypoplasia** | | | |
| 10 | M/34 | 0.0/0.0 | R, L: unclear subjective visual disturbance |
| 11 | M/35 | -0.1/(0.2) | L: ocular ischemia resulting from occlusion of the left arteriae carotides communis interna and externa |
| 12 | F/10 | 0.1/0.4 | R, L: hyperopia |
| 13 | M/47 | (0.4)/0.0 | R: contusio bulbi, central corneal erosio |
| 14 | M/52 | (0.2)/-0.1 | R: optic neuropathy and congestive papilla |
| 15 | M/25 | 0.0/0.1 | R, L: astigmatism |
| 16 | F/39 | (0.1)/ (0.1) | R, L: cataract, vitreous opacity |
| 17 | M/16 | 0.0/0.0 | R, L: papilla excavation without optic atrophy<br>L: cellophane maculopathy |
| 18 | M/53 | 0.0/(0.1) | R, L: myopia<br>L: pars plana vitrectomy after retinal detachment |
| 19 | F/6 | (0.5)/ (0.5) | R, L: hereditary motor sensitive neuropathy I, amblyopia, nystagmus; R: cellophane maculopathy |
| 20 | M/10 | 0.2/0.1 | R, L: macro-optic disc |
| 21 | F/37 | 0.0/0.0 | R, L: normal tension glaucoma, optic disc pit |
| 22 | M/9 | 0.1/0.1 | R, L: unclear subjective visual disturbance |

F, female; HFL, Henle fiber layer; L, left eye; M, male; n.d., not determined; R, right eye; BCVA, best-corrected visual acuity in logMAR. Visual acuity values in parenthesis were apparently influenced by extraretinal conditions.

RPE-Bruch's membrane interface to the internal limiting membrane) were measured. The thickness of inner retinal layers was measured between the internal limiting membrane and the outer margin of the OPL. In the foveola of control eyes, the horizontal layer of the Müller cell cone (MCC) forms the inner layer [34]. The thickness of outer retinal layers was measured between the outer margin of the OPL (and of the horizontal layer of the MCC in the foveola of control eyes) and the RPE-Bruch's membrane interface.

Data are given as means ± S.D. BCVA is shown as median ± interquartile range. Statistical analysis included calculation of significant differences and Spearman correlation coefficients ($r$), and was performed using GraphPad Prism 6.07 (GraphPad Software, San Diego, CA). Significant differences were evaluated with one-way ANOVA and Mann-Whitney $U$ test, and were accepted at $P<0.05$.

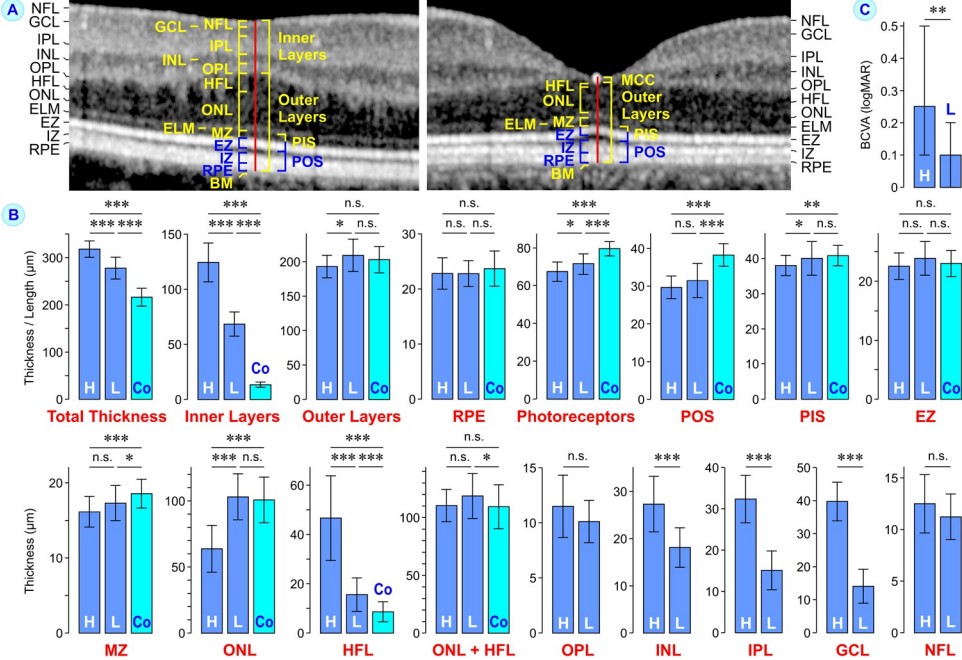

**Fig 3. Morphological parameters of the macular center of eyes with foveal hypoplasia. A.** The parameters were measured on SD-OCT images along a vertical line through the site of the lowest thickness of the inner retinal layers at the macular center of eyes with foveal hypoplasia (*left*) and the lowest thickness of the foveola in control eyes (*right*). In control eyes, the horizontal layer of the Müller cell cone (MCC) represents the inner layer of the foveal center. **B.** Mean ± S.D. total thickness and thicknesses of individual retinal layers at the macular/foveal center. Eyes with foveal hypoplasia were classified into two groups: high-grade (H) foveal hypoplasia with thick inner retinal layers at the macular center (n = 16) and low-grade (L) foveal hypoplasia with thinner inner retinal layers at the macular center (n = 25). As comparison, the thickness of the foveal center of control eyes (Co) with a regular fovea is shown (n = 75). **C.** Medians ± interquartile range of best-corrected visual acuity (BCVA) of subjects with foveal hypoplasia. *P<0.05; **P<0.01; ***P<0.001. n.s., not significant. BM, Bruch's membrane: ELM, external limiting membrane; EZ, ellipsoid zone; GCL, ganglion cell layer; HFL, Henle fiber layer; INL, inner nucler layer; IPL, inner plexiform layer; IZ, interdigitation zone; MZ, myoid zone; NFL, nerve fiber layer; ONL, outer nuclear layer; OPL, outer plexiform layer; PIS, photoreceptor inner segments; POS, photoreceptor outer segments; *r*, Spearman correlation coefficient; RPE, retinal pigment epithelium.

## Results

Eyes with foveal hypoplasia were classified into two groups: high-grade foveal hypoplasia characterized by thick inner retinal layers at the macular center (thickness above 100 μm; range 101–166 μm; n = 16 eyes of 9 subjects) and low-grade foveal hypoplasia with thinner inner retinal layers at the macular center (below 100 μm; range 48–97 μm; n = 25 eyes of 13 subjects). "Low-grade foveal hypoplasia" is comparable with grade 1 foveal hypoplasia of Thomas et al. [26] whereas "high-grade foveal hypoplasia" includes morphological features which were described by Thomas et al. [26] by grades 2 to 4 foveal hypoplasia. Two eyes of two subjects with high-grade foveal hypoplasia were not included in data analysis because there were no SD-OCT images of one eye and the other eye showed epiretinal membrane-mediated thickening of the macular tissue. One eye with low-grade foveal hypoplasia was excluded because of retinal deterioration associated with optic neuropathy.

### Morphological parameters of foveal hypoplasia

Horizontal SD-OCT scans through the macula of eyes with high-grade foveal hypoplasia showed a thick macular center and an absent or very shallow foveal pit; the inner retinal layers

(nerve fiber layer to OPL) traversed the macular center as continuous bands (Figs 1A–1C and 2A–2C). A clear fovea externa was absent. The increased thickness of the macular center resulted from thickening of inner, but not outer retinal layers as compared to the layer thicknesses of control eyes (Fig 3B). In eyes with low-grade foveal hypoplasia, which were characterized by a shallow foveal pit and the presence of a fovea externa (Fig 2D, 2Ea), the total thickness of the foveal center and the thickness of the central inner retinal layers were significantly ($P<0.001$) smaller than that of eyes with high-grade foveal hypoplasia, but thicker than that of control eyes (Fig 3B). On the other hand, the thickness of the central outer retinal layers was slightly decreased in eyes with high-grade foveal hypoplasia and not altered in eyes with low-grade foveal hypoplasia compared to control (Fig 3B).

The thickness of the central RPE was not different between eyes with foveal hypoplasia and control eyes (Fig 3B). Photoreceptors and POS in the macular center were significantly ($P<0.001$) shorter in eyes with foveal hypoplasia compared to control eyes (Fig 3B). The length of PIS was shorter in eyes with high-grade foveal hypoplasia (Fig 3B). Whereas the thickness of the EZ of central PIS was similar in all eye populations investigated, the thickness of MZ was smaller in eyes with foveal hypoplasia compared to control eyes (Fig 3B). The central ONL was significantly ($P<0.001$) thinner in eyes with high-grade foveal hypoplasia and not altered in thickness in eyes with low-grade foveal hypoplasia compared to control (Fig 3B). The central HFL was thicker in eyes with foveal hypoplasia compared to control eyes; the degree of HFL thickening depended on the grade of foveal hypoplasia (Fig 3B). The total thickness of both ONL and HFL in eyes with high-grade foveal hypoplasia was similar to that of control eyes while eyes with low-grade foveal hypoplasia showed a slightly increased thickness of both ONL and HFL compared to control (Fig 3B). The difference in the thickness of the central inner retinal layers between eyes with high-grade and low-grade foveal hypoplasia mainly resulted from different thicknesses of the inner nucler layer, inner plexiform layer, and GCL (Fig 3B).

Calculation of Spearman correlation coefficients revealed that in eyes with foveal hypoplasia the lengths of photoreceptors, POS, and PIS at the macular center were positively related to the thickness of the central ONL, and the lengths of photoreceptors and PIS were negatively related to the thickness of the inner retinal layers (Table 2). In contrast, there was no correlation between POS length and ONL thickness in the foveal center of control eyes (Table 3). There was no correlation between the lengths of POS and PIS in eyes with foveal hypoplasia (Table 2). The thickness of the central ONL in eyes with foveal hypoplasia was negatively and the thickness of the central HFL was positively correlated to the thickness of inner retinal layers (Table 2). There was a negative correlation between the thicknesses of the central ONL and HFL in eyes with foveal hypoplasia (Table 2) whereas the thicknesses of both layers were positively related in control eyes (Table 3).

Fig 4A and 4B shows mean schematic cross sections through the macular center of subjects with high- and low-grade foveal hypoplasia. As comparison, Fig 4C shows mean schematic cross sections through the nasal hemimeridians of control eyes. The sections in Fig 4A display the variability of the thicknesses of the central ONL and HFL in eyes with high-grade foveal hypoplasia despite similar thicknesses of the inner and outer retinal layers. There were varying degrees of central ONL thickening associated with a thinning of the central HFL (Fig 4A).

The thicknesses of the ONL and HFL, and the length of POS, were also measured at 0.5 and 1.0 mm nasal from the macular/foveal center on horizontal SD-OCT images; the former is the site of the thickest mean HFL and the latter is the site of the thickest tissue of the parafovea (Fig 4C). All eyes investigated had a thicker ONL at the macular/foveal center than at 1.0 mm nasal from the center (Fig 5A). In the mean, eyes with high-grade foveal hypoplasia had a 1.3-fold thicker ONL at the macular center compared to the ONL 1.0 mm nasal from the macular center. Eyes with low-grade foveal hypoplasia had a 2.6-fold thicker ONL in the foveal

**Table 2. Correlations between the total thickness and thicknesses of individual retinal layers at the macular center of eyes with foveal hypoplasia (n = 41).**

| Retinal Layer | Retinal Layer | r | Significance |
|---|---|---|---|
| Total thickness | Inner Retinal Layers | 0.740 | P<0.001 |
| | Outer Retinal Layers | 0.125 | n.s. |
| Inner Retinal Layers | Outer Retinal Layers | -0.466 | P<0.01 |
| RPE | POS | -0.166 | n.s. |
| PR Length | ONL Thickness | 0.557 | P<0.001 |
| | HFL Thickness | -0.136 | n.s. |
| | Inner Retinal Layers | -0.402 | P<0.01 |
| POS Length | PIS Length | 0.122 | n.s. |
| | ONL Thickness | 0.446 | P<0.01 |
| | HFL Thickness | -0.215 | n.s. |
| | Inner Retinal Layers | -0.273 | n.s. |
| PIS Length | ONL Thickness | 0.329 | P<0.05 |
| | HFL Thickness | -0.023 | n.s. |
| | Inner Retinal Layers | -0.344 | P<0.05 |
| ONL Thickness | HFL Thickness | -0.728 | P<0.001 |
| | Inner Retinal Layers | -0.718 | P<0.001 |
| HFL Thickness | Inner Retinal Layers | 0.671 | P<0.001 |

GCL, ganglion cell layer; HFL, Henle fiber layer; NFL, nerve fiber layer; n.s., not significant; ONL, outer nuclear layer; PIS, photoreceptor inner segments; POS, photoreceptor outer segments; PR, photoreceptor; r, Spearman correlation coefficient; RPE, retinal pigment epithelium.

center than in the parafovea which was significantly (P<0.001) lower than in control eyes (3.2-fold). The paracentral ONL was thickest in eyes with high-grade foveal hypoplasia and thinnest in control eyes (Fig 5A). There was a significant (P<0.01) correlation between the thicknesses of the ONL at the macular center and 1.0 mm nasal from the center in eyes with high-grade foveal hypoplasia (Table 3). Such correlations were not found for eyes with low-grade foveal hypoplasia and control eyes (Table 3).

**Table 3. Correlations between the length of photoreceptor outer segments (POS) and the thickness of the outer nuclear layer (ONL) at the macular/foveal center and 1.0 mm nasal from the center for eyes with high-grade (H; n = 16) and low-grade (L) foveal hypoplasia (n = 25), and control eyes (Co; n = 75).**

| Parameter | Parameter | Eyes | r | Significance |
|---|---|---|---|---|
| Central POS | Paracentral POS | H | 0.636 | P<0.01 |
| | | L | 0.123 | n.s. |
| | | Co | 0.217 | n.s. |
| Central ONL | Paracentral ONL | H | 0.722 | P<0.01 |
| | | L | 0.187 | n.s. |
| | | Co | 0.054 | n.s. |
| Paracentral POS | Paracentral ONL | H | -0.034 | n.s. |
| | | L | 0.044 | n.s. |
| | | Co | 0.099 | n.s. |
| Central POS | Central ONL | Co | -0.146 | n.s. |
| Central ONL | Central HFL | Co | 0.295 | P<0.05 |

HFL, Henle fiber layer; n.s., not significant; r, Spearman correlation coefficient.

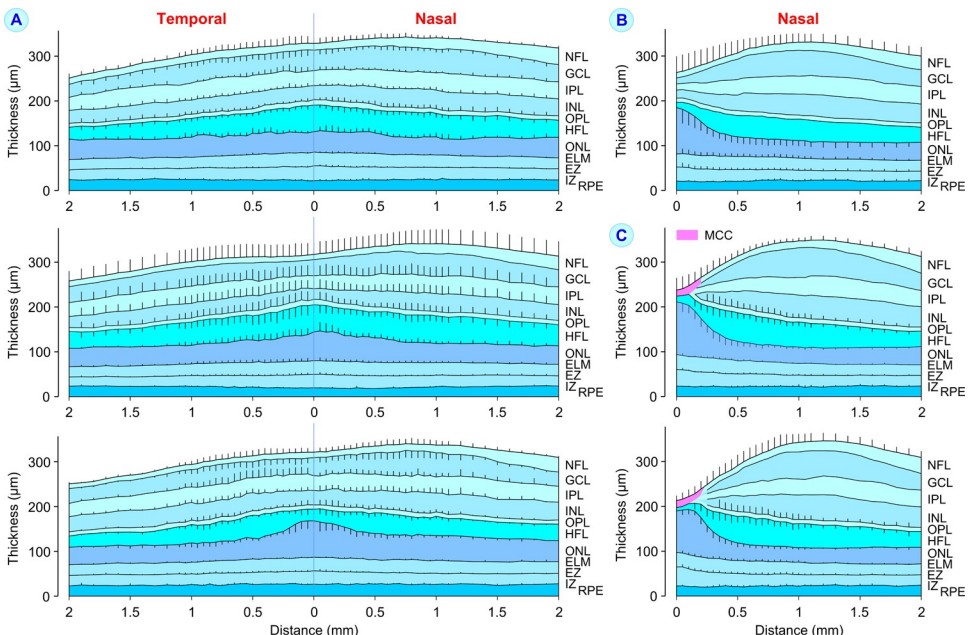

**Fig 4.** Schematic horizontal cross sections through the macula of eyes with high-grade foveal hypoplasia (**A**) and through the nasal hemimeridians of eyes with low-grade foveal hypoplasia (**B**) and eyes of control subjects (**C**). Retinal layer thickness in the central macula was measured in dependence on the distance to the macular center; the *black lines* show mean ± S.D. retinal layer borders. The images in **A** show sections through the macula of eyes with large (*above* and *middle*) and small hyporeflective zones in the central Henle fiber layer (HFL; *below*). The images in **C** show sections through foveas with the thickest outer nuclear layer (ONL) in the center of the foveola (*above*) and in the peripheral foveola (*below*). The sections were calculated from data of 4 eyes of subjects 1 and 2 (**A**, *above*), 5 eyes of subjects 3, 4, and 5 (**A**, *middle*), 3 eyes of subjects 6 and 9 (**A**, *below*), 6 eyes of subjects 10, 13, 19, 21, and 22 (**B**), 5 eyes of 5 control subjects (**C**, *above*), and 4 eyes of 4 further control subjects (**C**, *below*). The levels of external limiting membrane (ELM), ellipsoid zone (EZ), and interdigitation zone (IZ) lines were measured at the middle of the lines. GCL, ganglion cell layer; INL, inner nuclear layer; IPL, inner plexiform layer; MCC, Müller cell cone; NFL, nerve fiber layer; OPL, outer plexiform layer; RPE, retinal pigment epithelium.

The mean thickness of the HFL was not different between the macular center and paracentral areas in eyes with high-grade foveal hypoplasia (Fig 5B). The thickness of the HFL at 1.0 mm nasal from the macular center was not different between the three eye populations investigated (Fig 5B). The HFL at 0.5 mm nasal from the macular center was thicker than the more peripheral HFL in control eyes; such a thickening of the paracentral HFL at this site was not observed in eyes with foveal hypoplasia (Fig 5B). The total thickness of both ONL and HFL at 1.0 mm nasal from the macular center was highest in eyes with high-grade foveal hypoplasia and lowest in control eyes (Fig 5C).

All eyes investigated had longer POS at the macular/foveal center than in paracentral areas; the difference was lowest in eyes with high-grade foveal hypoplasia (Fig 5D). The length of paracentral POS was not different between the three eye populations (Fig 5D). In the mean, eyes with high-grade foveal hypoplasia had 1.26-fold longer POS at the macular center compared to the POS 1.0 mm nasal from the center; this ratio was significantly ($P<0.001$) lower than that of control eyes (1.57-fold; Fig 5D). Eyes with low-grade foveal hypoplasia had 1.31-fold longer POS in the foveal center than in the parafovea ($P<0.001$ compared to control eyes). The lengths of central and paracentral POS were only correlated in eyes with high-grade foveal hypoplasia (Table 3). There was no correlation between the POS length and ONL thickness in the paracentral area (Table 3).

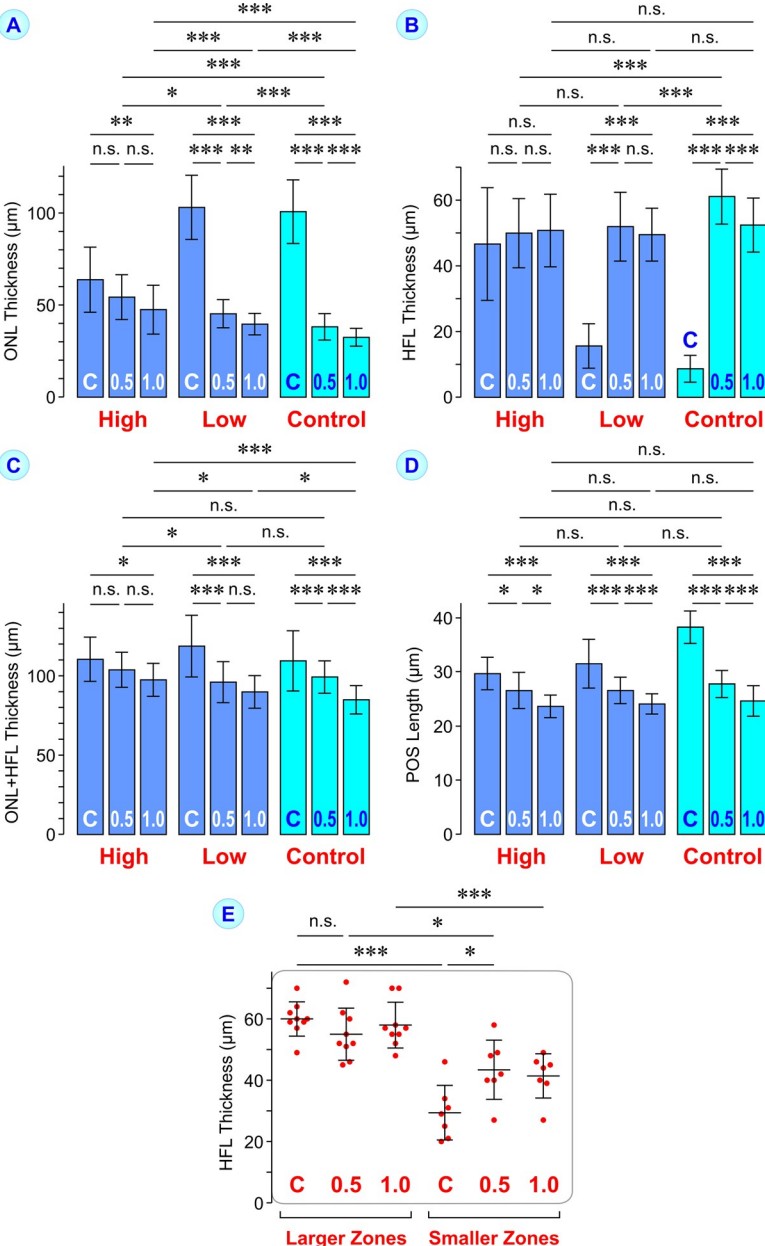

**Fig 5.** Comparison of the mean ± S.D. thicknesses of the outer nuclear layer (ONL; **A**) and Henle fiber layer (HFL; **B**), the total thickness of both ONL and HFL (**C**), and the length of photoreceptor outer segments (POS; **D**) between the macular/foveal center (C) and at 0.5 and 1.0 mm nasal from the center. The data were measured on horizontal SD-OCT images of eyes with high- (n = 16) and low-grade foveal hypoplasia (n = 25), and of control eyes (n = 75). **E.** Mean ± S.D. thickness of the HFL in the macular/foveal center (C) and at 0.5 and 1.0 mm nasal from the center. The data were measured on horizontal SD-OCT images of eyes with high-grade foveal hypoplasia and larger (n = 9) and smaller hyporeflective zones in the HFL (n = 7), respectively. *$P<0.05$; **$P<0.01$; ***$P<0.001$. n.s., not significant.

## Visual acuity

High-grade foveal hypoplasia was associated with a significant ($P<0.01$) lower mean BCVA compared to low-grade foveal hypoplasia (Fig 3C). BCVA of subjects with foveal hypoplasia was negatively correlated to the thickness of inner retinal layers and positively correlated to the

**Table 4. Correlations between best-corrected visual acuity (BCVA) and morphological parameters of the macular center of subjects with foveal hypoplasia.**

|  | Retinal Layer | r | Significance |
|---|---|---|---|
| BCVA | Total thickness | -0.263 | n.s. |
|  | Inner Retinal Layers | -0.349 | $P<0.05$ |
|  | Outer Retinal Layers | 0.464 | $P<0.01$ |
|  | POS Length | 0.340 | $P<0.05$ |
|  | PIS Length | 0.261 | n.s. |
|  | ONL | 0.367 | $P<0.05$ |
|  | Ratio | 0.429 | $P<0.01$ |
|  | HFL | -0.224 | n.s. |
|  | INL | -0.464 | $P<0.01$ |
|  | IPL | -0.270 | $P<0.05$ |
|  | GCL | -0.293 | $P<0.05$ |

Ratio means the ratio between the thicknesses of the central outer nuclear layer (ONL) and the ONL at 1.0 mm nasal from the macular center. GCL, ganglion cell layer; HFL, Henle fiber layer; INL, inner nuclear layer; IPL, inner plexiform layer; n.s., not significant; PIS, photoreceptor inner segments POS, photoreceptor outer segments; *r*, Spearman correlation coefficient.

thickness of outer retinal layers at the macular center (Table 4). BCVA of subjects with foveal hypoplasia was positively correlated to the length of POS, but not the length of PIS (Table 4). There were also positive correlations between BCVA and the thickness of the central ONL as well as the ratio between the thicknesses of the central and paracentral ONL (Table 4). BCVA was negatively correlated to the thicknesses of individual inner retinal layers (Table 4).

## Hyporeflective zones in the HFL

We found on SD-OCT images that the HFL in the macula of eyes with high-grade foveal hypoplasia contained hyporeflective zones which were not present in the HFL of eyes with low-grade foveal hypoplasia and control eyes. The hyporeflective zones in eyes with high-grade foveal hypoplasia caused a striped appearance of the HFL with alternating more or less vertical hypo- and hyperreflective bands (Figs 1Ab–1Cb and 2A–2C). The hyperreflective bands likely represent Henle fiber bundles which draw through the HFL.

The size of the hyporeflective zones in the HFL varied among eyes with high-grade foveal hypoplasia. The sections in Fig 4A were calculated from data of eyes with large hyporeflective zones in the HFL (*above* and *middle*) and of eyes with smaller hyporeflective zones in the HFL (*below*). Apparently, the size of the hyporeflective zones in the HFL varied in dependence on the thickness of the central ONL. Furthermore, the shape of the hyporeflective zones varied. Many zones extended through the entire HFL while some spaces filled only a part of this layer (Figs 1Ab–1Cb and 2A, 2B). In addition, the spatial arrangement of the Henle fiber bundles was different; many bundles drawed vertically through the HFL while other bundles were obliquely arranged (Figs 1Ab–1Cb and 2A, 2B).

Because many hyporeflective zones extended through the entire HFL, the thickness of the HFL should be different between eyes with high-grade foveal hypoplasia and larger and smaller hyporeflective zones in the HFL. Indeed, the central and paracentral HFL were significantly thicker in eyes with larger hyporeflective zones compared to eyes with smaller zones (Fig 5E). The central HFL of eyes with larger hyporeflective zones had a thickness of 49–70 μm whereas that of eyes with smaller zones was 20–46 μm (Fig 5E).

### OCTA images

OCTA images of eyes with high-grade foveal hypoplasia showed the absence of a foveal avascular zone with retinal vessels crossing the entire central macula at the levels of the superficial and deep vascular plexus (Fig 1Ac). OCTA images of an eye with low-grade foveal hypoplasia demonstrated a foveal avascular zone at the level of the deep vascular plexus but not at the superficial vascular plexus (Fig 2Eb). OCTA images recorded at the level of the HFL of eyes with high-grade foveal hypoplasia showed (in addition to shadows produced by overlying larger vessels) concentric rings of different reflectivity around the macular center (Fig 1Ac, 1Bc, 1Cc). We did not observe similar rings on fundus photographs of eyes with high-grade foveal hypoplasia (Figs 1Bd and 2C) or on OCTA images of eyes with low-grade foveal hypoplasia and control eyes.

## Discussion

Underdevelopment of the foveal pit results in a macular center which is thicker than the foveal center of normal control eyes (Fig 3B). The total thickness of the macular center of eyes with foveal hypoplasia showed a correlation with the thickness of the inner retinal layers but not the thickness of outer layers (Table 2).

### Avascular zone in eyes with foveal hypoplasia

It was described that eyes with foveal hypoplasia lack a central avascular zone at the level of the superficial vascular plexus while the deep vascular plexus is absent to varying degrees, with most persistent vessels in eyes with high-grade foveal hypoplasia [35]. We found using OCTA imaging that eyes with high-grade foveal hypoplasia lack central avascular zones at the levels of both the superficial and deep vascular plexus (Fig 1Ac). An eye with low-grade foveal hypoplasia showed the presence of an avascular zone at the level of the deep vascular plexus and the absence of an avascular zone at the level of the superficial vascular plexus (Fig 2Eb). Pakzad-Vaezi et al. [35] hypothesized that the presence of an avascular zone at the level of the deep vascular plexus in eyes with low-grade foveal hypoplasia allows the developmental thickening of the central ONL and the formation of the fovea externa whereas the lack of an avascular zone at this level in eyes with high-grade foveal hypoplasia inhibits these processes.

### Photoreceptors in eyes with foveal hypoplasia

Maximal visual acuity depends on the peak photoreceptor density in the central fovea [36]. We found that the central photoreceptors and POS were shorter in eyes with foveal hypoplasia compared to control eyes (Fig 3B). The length of POS was reduced in the macular center, but not in paracentral areas, of eyes with foveal hypoplasia compared to control (Fig 5D). Because the maximal POS length is positively correlated to the central photoreceptor packing density in subjects with albinism and foveal hypoplasia [27, 37], the data may indicate that the central photoreceptor density is smaller in eyes with foveal hypoplasia compared to control eyes. The present data are in accordance with previous studies which showed that eyes with high-grade foveal hypoplasia display a reduced length of central POS and lack a regular fovea externa [26, 38]. However, the small difference between the lengths of central and paracentral POS (Fig 5D) suggests that a minimal fovea externa, which develops by POS lengthening, is also formed in eyes with high-grade foveal hypoplasia.

 We found in agreement with a previous study [32] that the lengths of central photoreceptors, POS, and PIS were positively correlated to the central ONL thickness in eyes with foveal hypoplasia (Table 2). On the other hand, there was no correlation between the ONL thickness

and POS length in the foveal center of control eyes (Table 3), suggesting that the developmental thickening of the ONL proceeds partly independent on the centripetal displacement of photoreceptor segments. The developmental displacement of the photoreceptor segments likely occurs at the ELM while the displacement of the photoreceptor cell somata occurs in the ONL [4]. In many control subjects, the centralmost ONL contains a reduced number of photoreceptor cell somata compared to the ONL of the more peripheral foveola, probably to improve light transfer through the central tissue [4, 39]. This means that the somata of many central photoreceptor cells remain peripherally displaced; there are relatively long outer fibers of the photoreceptor cells which connect the receptor segments at the ELM with the somata in the ONL. The more centripetal displacement of the receptor segments compared to the cell somata supports the assumption that the displacement of the segments proceeds partly independent from that of the somata.

Because of the persistance of inner retinal layers at the macular center, it is likely that eyes with high-grade foveal hypoplasia do not contain a regular MCC which forms the inner layer of the foveola in control eyes [34]. The fovea externa, i.e., the cone-like arrangement of the elongated photoreceptor segments in the foveola, was proposed to be formed by the elongation of POS, a vertical contraction of the stalk of the MCC, and an oblique pulling of the Henle fibers resulting from the centrifugal displacement of the inner foveal layers [4]. All these factors are absent or reduced in eyes with high-grade foveal hypoplasia which may explain the absence of a regular fovea externa.

## Visual acuity of subjects with foveal hypoplasia

In subjects with albinism and foveal hypoplasia, the central POS length was described to be the strongest predictor of visual acuity [26, 40]. We found in agreement with previous studies [24, 30] that BCVA depended on the grade of hypoplasia (Fig 3C). BCVA of subjects with foveal hypoplasia was negatively correlated to the thickness of the central inner retinal layers and positively correlated to the thicknesses of the central outer retinal layers and ONL, and the length of POS (Table 4). Because the length of POS is proportional to the photoreceptor packing density in subjects with albinism and foveal hypoplasia [37], the data may suggest that BCVA of subjects with foveal hypoplasia increases with a higher central photoreceptor density. However, because foveal hypoplasia is often associated with alterations in extraretinal conditions which influence visual acuity, the difference in mean BCVA (Fig 3C) may result from both retinal and extraretinal differences between subjects with high- and low-grade foveal hypoplasia. Two retinal factors which may contribute to some degree to the variation of visual acuity of subjects with foveal hypoplasia are the thickness of the inner retinal layers and the POS length at the macular center.

## ONL and HFL in eyes with foveal hypoplasia

We found in eyes with foveal hypoplasia that the ratio between the thicknesses of the ONL at the macular center and 1.0 mm nasal from the center was greater than 1.0 (Fig 5A). At birth, the foveal ONL is a monolayer of photoreceptor cell somata whereas the ONL in the parafovea contains 2–3 rows of stacked somata [41]. Therefore, the ratio of greater than 1.0 suggests that a centripetal displacement and central accumulation of photoreceptor cell somata occurred in eyes with foveal hypoplasia during postnatal development. The lower thickness of the central ONL in eyes with high-grade compared to low-grade foveal hypoplasia (Fig 3B) may suggest that less photoreceptor cell somata are displaced into the macular center of these eyes. On the other hand, we also found that the paracentral ONL is thicker in eyes with high-grade foveal hypoplasia compared to control eyes (Fig 5A). In eyes with low-grade foveal hypoplasia, the thickness of the central ONL is similar to that in control eyes (Fig 3B) while the paracentral

ONL is thicker (Fig 5A). The paracentral ONL is thicker in eyes with high-grade compared to low-grade foveal hypoplasia (Fig 5A). There was a correlation between the thicknesses of the central and paracentral ONL in eyes with high-grade foveal hypoplasia which was not found in eyes with low-grade foveal hypoplasia and control eyes (Table 3). The reason for these differences is unclear. The data suggest that the postnatal centriptal displacement of photoreceptor cell somata proceeds partly different in the three eye populations investigated.

The correlation between the thicknesses of the central and paracentral ONL in eyes with high-grade foveal hypoplasia (Table 3) and the different thicknesses of the central and paracentral ONL between the three eye populations (Fig 5A) might be explained with the assumption that the centripetal displacement of photoreceptor cell somata during development and the accumulation of the somata in the foveal center are mediated by two different processes. The first results in a thickening of the ONL at both the central and paracentral macula. The second results in a thickening of the central ONL which is associated with a thinning of the paracentral ONL. The first process may proceed similar in all eye populations investigated whereas the second process proceeds only in a rudimentary fashion in eyes with high-grade foveal hypoplasia, possibly due to the presence of blood vessels at the deep vascular plexus of the macular center [35]; the latter may explain the lower thickness of the central ONL and the higher thickness of the paracentral ONL in eyes with high-grade foveal hypoplasia compared to control eyes (Fig 5A). The absence of a correlation between the thicknesses of the central and paracentral ONL in eyes with low-grade foveal hypoplasia and control eyes (Table 3) might be explained with the assumption that the thickness of the central ONL is determined by two different processes, centripetal displacement of photoreceptor cell somata and somata displacement from the paracentral area into the foveal center which lowers the amount of somata in the paracentral ONL. The presence of a correlation between the thicknesses of the central and paracentral ONL in eyes with high-grade foveal hypoplasia (Table 3) might be explained with the assumption that the thicknesses of both central and paracentral ONL are only dependent on one process which proceeds similar in central and paracentral areas.

The horizontal extension of Henle fibers reflects the developmental shift between inner and outer retinal layers. The Henle fiber bundles which surround the hyporeflective zones in the HFL of eyes with high-grade foveal hypoplasia (Figs 1Ab–1Cb and 2A–2C) show that there is a short horizontal extension of Henle fibers in these eyes; the fiber bundles draw more or less vertically through the HFL (Fig 6A). In contrast, Henle fibers in control eyes are long (with an average peak length of about 550 μm [42]) and draw rather horizontally through the HFL (Fig 6C); Henle fibers in eyes with low-grade foveal hypoplasia may have intermediate length and position (Fig 6B). The differences can be explained with the different positions of bipolar cells and photoreceptor cell somata in the three eye populations. In eyes with high-grade foveal hypoplasia, bipolar cells are not centrifugally displaced and photoreceptor cell somata are not enriched in the macular center; both may explain the short horizontal extension of Henle fibers in these eyes (Fig 6D). In eyes with low-grade foveal hypoplasia, only a part of central bipolar cells are centrifugally displaced and photoreceptor cell somata are enriched in the macular center; both may explain the lengthening and oblique arrangement of Henle fibers in these eyes (Fig 6E). In control eyes, all bipolar cells are displaced from the foveal center and photoreceptor cell somata are enriched in the foveola; this explains the length and spatial arrangement of Henle fibers (Fig 6F).

## Hyporeflective zones in the HFL

We found on SD-OCT images of eyes with high-grade foveal hypoplasia that the HFL contained hyporeflective zones (Figs 1Ab–1Cb and 2A–2C) which were not present in the HFL of

eyes with low-grade foveal hypoplasia (Fig 2D, 2Ea) and control eyes. We assume that the hyporeflective zones in the HFL represent cystoid spaces which are surrounded by Henle fiber bundles. The presence of cystoid spaces may be the reason for the different reflectivities which cause the concentric rings around the macular center at the HFL of eyes with high-grade foveal hypoplasia (Fig 1Ac, 1Bc, 1Cc). A colocalization of concentric macular rings with alternating hyporeflective and hyperreflective vertical bands in the HFL on OCT images of eyes with foveal hypoplasia was recently described (Ramtohul et al., 2020). We found that the HFL in some eyes with low-grade foveal hypoplasia and control eyes may exhibit a striped appearance with alternating vertical medium- and higher reflective bands (e.g., lowest image of Fig 2D); however, hyporeflective zones in the HFL were missing. Because a higher reflectivity of the HFL can be achieved by alteration of the direction of the incident light [43, 44], the increased reflectivity of the HFL in eyes with low-grade foveal hypoplasia might be caused by an alteration the position of Henle fibers resulting from underdevelopment of the fovea and/or tractional distortion of the foveal tissue. Similarly, the hyperreflectivity of the Henle fiber bundles which surround the cystoid spaces in the HFL of eyes with high-grade foveal hypoplasia might be caused by an altered position of Henle fibers to the incident light. Gliosis of Müller cell processes, which surround the photoreceptor cell axons [34] and which is associated with cell process hypertrophy and upregulation of glial intermediate filaments [45, 46] that both alter the optical properties of the cells, may contribute to the increased reflectivity of erected Henle fiber bundles.

There is a negative relation between the thicknesses of the central ONL and HFL in eyes with foveal hypoplasia (Table 2). Because the total thickness of both ONL and HFL is not different between eyes with high-grade foveal hypoplasia and control eyes (Fig 3B), a decrease in the thickness of the central ONL results in increased thickness of the central HFL. A reduced thickness of the central ONL (Fig 3B) may reflect a reduced photoreceptor cell density at the macular center of eyes with high-grade foveal hypoplasia [32]. This means that the thickened central HFL in these eyes (Fig 3B) contains a lower density of Henle fibers compared to eyes with low-grade hypoplasia. These factors (enlarged thickness of the central HFL due to the lower amount of photoreceptor cell somata in the central ONL and the decreased density of short Henle fibers) may be reasons why the HFL of eyes with high-grade foveal hypoplasia contains cystoid spaces.

The formation of cystoid spaces in the HFL may be supported by the histological feature of this layer. The HFL is a layer with a low mechanical cohesion because it is composed of Henle fibers which are not connected and which can be shifted or even erected, e.g., in eyes with anterior traction to the macula or due to intraretinal fluid accumulation resulting in foveoschisis or edematous cystoid spaces [46–48]. Erection of Henle fibers results in a thickening of the HFL and a large increase of the extracellular space volume within the HFL. Cystoid macular edema associated with increased foveal thickness is common in preterm infants; however, edematous cysts are predominantly located in the inner nuclear layer and not in the HFL [49].

## Further implications for understanding of foveal underdevelopment

Foveal hypoplasia is a macular disorder which results from underdevelopment of the fovea [18]. The mechanisms which cause the lack or reduction of foveal pit and fovea externa formation are not fully understood. Both genetic and nongenetic factors like prematurity of birth can affect the foveal development.

In eyes with high-grade foveal hypoplasia, both the centrifugal displacement of the inner retinal layers and the displacement of photoreceptor cell somata into the macular center are reduced, resulting in a low horizontal extension of Henle fibers; the short Henle fibers remain

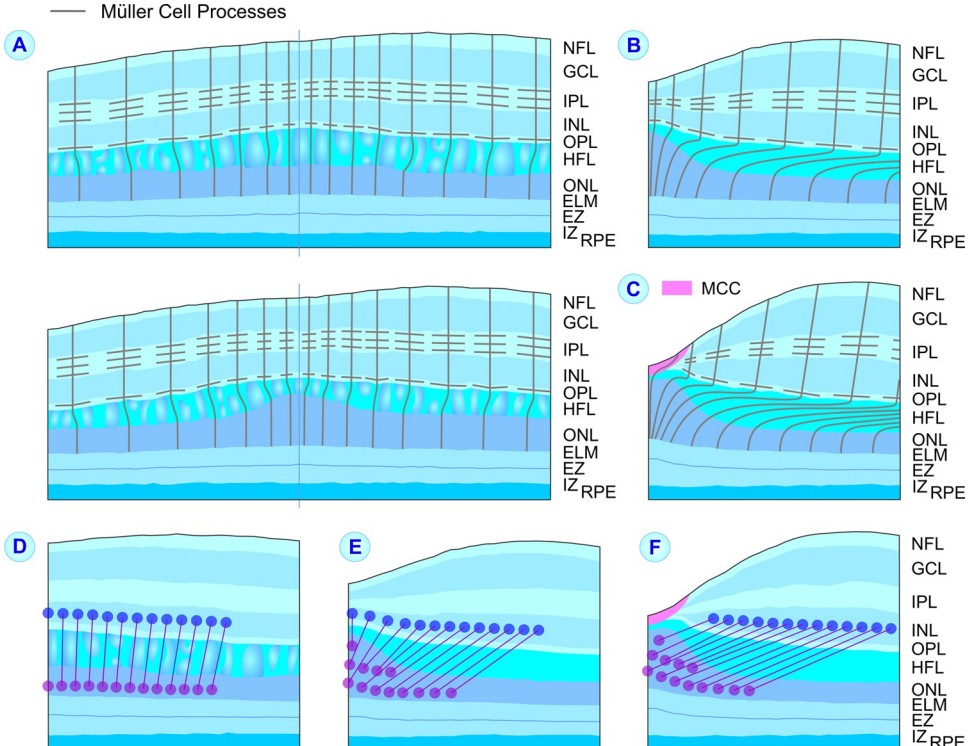

**Fig 6.** Proposed spatial arrangement of Müller cells (**A–C**) and spatial relation between photoreceptor cell somata (*purple circles*) and bipolar cells (*blue circles*; **D–F**) at the macular center of eyes with high-grade (**A, D**) and low-grade foveal hypoplasia (**B, E**), and control eyes (**C, F**). **A–C.** Because Henle fibers are composed of photoreceptor cell axons which are surrounded by Müller cell processes, the spatial arrangement of Henle fibers indicates the position of the outer Müller cell processes in the Henle fiber layer (HFL). The orientation and horizontal extension of Henle fibers differ between the three eye populations. In eyes with high-grade foveal hypoplasia, short Henle fibers traverse more or less vertically the HFL. In eyes with low-grade foveal hypoplasia, longer Henle fibers traverse obliquely the HFL. In control eyes, long Henle fibers traverse rather horizontally the HFL. There is no oblique arrangement of the Müller cell processes in the inner layers of eyes with high-grade foveal hypoplasia because there is no widening of the macular tissue by contraction of the astrocytic network around the macular center during development. **D–F.** The different magnitudes of the developmental displacements of the inner retinal layers and photoreceptor cells result in the formation of longer Henle fibers in control eyes compared to eyes with foveal hypoplasia. The levels of external limiting membrane (ELM), ellipsoid zone (EZ), and interdigitation zone (IZ) lines were measured at the middle of the lines. GCL, ganglion cell layer; INL, inner nuclear layer; IPL, inner plexiform layer; MCC, Müller cell cone; NFL, nerve fiber layer; n.s., not significant; ONL, outer nuclear layer; OPL, outer plexiform layer; RPE, retinal pigment epithelium.

more or less erected (Fig 6A and 6D). However, it is unclear why eyes with high-grade foveal hypoplasia display a relatively thick HFL (Figs 3B and 4A) although the developmental displacements, which normally form the Henle fibers, are reduced. We found in agreement with a previous study [32] that the total thickness of the ONL and HFL at the macular center is similar between eyes with high-grade foveal hypoplasia and normal control eyes (Fig 3B). This finding may suggest that the total thickness of the ONL and HFL is developmentally preserved and depends on other factors than the thickness of the inner foveal layers.

We assume that the distance between the central OPL and ELM is preserved by Müller cells during development in all eye populations investigated. The space between the central OPL and ELM is developmentally filled by displaced photoreceptor cell somata and Henle fibers. In dependence on the amount of displaced photoreceptor cell somata, the thickness of the remaining space (HFL) varies. This assumption may partly explain the negative correlation between the thicknesses of the ONL and HFL at the macular center of eyes with foveal

hypoplasia (Table 2) whereas in control eyes, both thicknesses are positively correlated (Table 3). Because relatively few photoreceptor cell somata are displaced into the macular center in eyes with high-grade foveal hypoplasia, the central ONL is relatively thin and the central HFL is thick (Fig 3B).

We assume that cystoid spaces in the HFL result from the thin ONL which is associated with a thick HFL because the thickness of both ONL and HFL is not altered in eyes with foveal hypoplasia compared to control eyes. The central OPL in some eyes with high-grade foveal hypoplasia has rather an angular than a circular shape and forms a cone with the tip at the macular center (Figs 1C, 2B, and 4A). This may suggest that centralmost Müller cells exert an anterior traction, possibly to preserve the distance between the central OPL and ELM. Anterior traction may also partially explain the finding that the total thickness of both the ONL and HFL in eyes with low-grade foveal hypoplasia is slightly thicker than in control eyes (Fig 3B). It is unclear why the distance between the central OPL and ELM is preserved during development; one possibility is that it occurs for optical reasons.

We found that the macular center of eyes with high- and low-grade foveal hypoplasia differ in various morphological parameters, e.g., the total thickness, the thicknesses of individual inner and outer retinal layers, and the photoreceptor length (Fig 3B). However, because of large variations of these morphological parameters, there were continuous transitions of various parameters between eyes with high- and low-grade foveal hypoplasia. We assume that the morphological differences between eyes with different grades of foveal hypoplasia result from varying amounts of astrocytes which migrate into the macular center during development; the extent of astrocyte migration is likely regulated by the amount of migration-inhibitory factors released from central Müller cells [4]. In developing eyes with high-grade foveal hypoplasia, a high amount of astrocytes (similar as in more peripheral retinal areas) migrate into the macular center whereas in eyes with lower grades of foveal hypoplasia, smaller amounts of astrocytes invade the macular center. Therefore, contraction of the astrocytic network around the macular center does not produce a centrifugal displacement of the inner retinal layers and a foveal pit in eyes with high-grade foveal hypoplasia; lower amounts of astrocytes in the macular center decrease the resistance for a displacement of astrocytes during contraction of the astrocytic network and thus increasing amounts of ganglions cells are centrifugally displaced, resulting in the formation of shallow foveal pits. Further research is necessary to understand the mechanisms of foveal underdevelopment in eyes with different grades of foveal hypoplasia.

## Supporting information

**S1 Data.**
(XLSX)

## Author Contributions

**Conceptualization:** Andreas Bringmann, Focke Ziemssen.

**Data curation:** Andreas Bringmann, Thomas Barth.

**Formal analysis:** Andreas Bringmann.

**Methodology:** Andreas Bringmann.

**Supervision:** Focke Ziemssen.

**Visualization:** Andreas Bringmann.

**Writing – original draft:** Andreas Bringmann.

**Writing – review & editing:** Thomas Barth, Focke Ziemssen.

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
