## [Decision Letter · Decision Letter 0]

23 Feb 2022

PONE-D-21-37391Morphology of foveal hypoplasia: Hyporeflective zones in the Henle fiber layer of eyes with high-grade foveal hypoplasia may represent cystoid spacesPLOS ONE

Dear Dr. Bringmann,

Thank you for submitting your manuscript to PLOS ONE. After careful consideration, we feel that it has merit but does not fully meet PLOS ONE’s publication criteria as it currently stands. Therefore, we invite you to submit a revised version of the manuscript that addresses the points raised during the review process.

Please address the in depth comments and concers raised by the reviewer. Your revised paper will be re-sent for review. 

We look forward to receiving your revised manuscript.

Kind regards,

Tudor C. Badea, M.D., M.A., Ph.D.

Academic Editor

PLOS ONE

Journal Requirements:

2. We note that you have referenced (ie. Bewick et al. [5]) which has currently not yet been accepted for publication. Please remove this from your References and amend this to state in the body of your manuscript: (ie “Bewick et al. [Unpublished]”) as detailed online in our guide for authors

Reviewers' comments:

Reviewer's Responses to Questions

**Comments to the Author**

1. Is the manuscript technically sound, and do the data support the conclusions?

Reviewer #1: Partly

2. Has the statistical analysis been performed appropriately and rigorously? 

Reviewer #1: No

3. Have the authors made all data underlying the findings in their manuscript fully available?

Reviewer #1: Yes

4. Is the manuscript presented in an intelligible fashion and written in standard English?

Reviewer #1: Yes

5. Review Comments to the Author

Reviewer #1: This manuscript provides detailed analyses on the OCT morphology in eyes with foveal hypoplasia. The hyporeflective changes in the Henle fiber layer found in this study are seemingly identical with those have been reported elsewhere (ref #30, this needs to be mentioned in the text). The authors have yet provided evidence that the hyporeflective zones represent cystoid spaces (line 595). It is better to change the title along with what the authors have found (not assumption). In relation to the hyporeflective zones, please answer: 1) if “the presence of cystoid spaces may be the reason for the different reflectivities which cause the concentric rings (line 597)”, the hyporeflective zones can be correspond to the area of cystoid space in the OCTA images. Did you find that? 2) the hyporeflective sings appear irregularly scattered while the reflection of the concentric rings showed regularity. How do you explain the difference?

There are additional comments:

1) For statistical analyses (including mean) BCVA should be converted to LogMAR.

2) It seems that the OCT image of the 1st low in Fig 2D seems to contain hyporeflective bands (right?).

3) I do not understand the merit of the present classification of foveal hypoplasia: a cut off value of 100 um of the thickness of the inner retina. How about using the earlier grading systems based on refs. #26 and #27? According to Fig 2, high and low grades can be grade 1 and 2, respectively.

4) Although Table 1 shows smaller and larger hyporeflective zones as a category, the exact extent has not been shown. Please add these values in Table 1. All OCT images including Figure 2 D should be referrable to Patient IDs. Please add them in the legends (figure reference in Table 1 is unnecessary).

5) Determining of the width of HFL at the foveal center is challenging in eyes with foveal pits because of the direction of the incident light: the thickness of HFL in eyes with low grade foveal hypoplasia and control seems inaccurate. The authors need to provide how the measurement was obtained accurately. Line 210: “the thickness of individual retinal layers was manually measured using the Heidelberg” is not enough (where is the border between HFL and ONL in Figure 3A right?).

6. PLOS authors have the option to publish the peer review history of their article (what does this mean?). If published, this will include your full peer review and any attached files.

Reviewer #1: No

---

## [Author Response · Author response to Decision Letter 0]

11 Mar 2022

Reviewer #1: 

Thank you very much for your helpful comments to improve the manuscript! 

 In the meantime, we realized that we made a mistake in the measurement of the length of the photoreceptor segments in the first version of the manuscript (we measured at the middle of the IZ, EZ, and ELM lines which is not right). Therefore, we remeasured the lengths of photoreceptors and photoreceptor inner and outer segments, as well as the thicknesses of the myoid and ellipsoid zones. The distances which we measured for the new version of the manuscript are shown in Figure 3A. We altered parts of the manuscript because we obtained new results regarding the photoreceptor length. The changes in the manuscript are marked in yellow. 

2. Has the statistical analysis been performed appropriately and rigorously? - Reviewer #1: No

It would be helpful for us if you can explain more precisely how the statistical analysis should be done. We calculated significant differences and Spearman correlation coefficients which is to our opinion adequate to support our conclusions. However, we are open for suggestions to improve statistical analysis. 

5. This manuscript provides detailed analyses on the OCT morphology in eyes with foveal hypoplasia. The hyporeflective changes in the Henle fiber layer found in this study are seemingly identical with those have been reported elsewhere (ref #30, this needs to be mentioned in the text). 

We included the following sentence in the Discussion section: "A colocalization of concentric macular rings with alternating hyporeflective and hyperreflective vertical bands in the HFL on OCT images of eyes with foveal hypoplasia was recently described [30]." 

 We mean that our data are only partly identical with those reported by Ramtohul et al. [30]. Ramtohul et al. [30] described the presence of alternating hyporeflective and hyperreflective vertical bands on OCT images of the HFL, but not the presence of hyporeflective zones. As we mentioned in the first version of the manuscript, we found alternating hyporeflective and hyperreflective bands in the HFL on SD-OCT images of some eyes with low-grade foveal hypoplasia (e.g., lowest image of Figure 2D) and control eyes; however, we did not observe concentric macular rings on OCTA images of these eyes. Therefore, to our opinion, alternating hyporeflective and hyperreflective vertical bands in the HFL alone are rather unlikely to reflect the cause of the concentric macular rings, at least in these eyes. This issue needs further investigations. 

The authors have yet provided evidence that the hyporeflective zones represent cystoid spaces (line 595). It is better to change the title along with what the authors have found (not assumption). 

Yes, we assume that the hyporeflective zones represent cystoid spaces; however, we have no direct evidence for this assumption. One indication is the hyporeflectivity which is similar to that of fluid-filled cysts in eyes with tractional foveal disorders. We shortened the title of the manuscript. 

In relation to the hyporeflective zones, please answer: 1) if “the presence of cystoid spaces may be the reason for the different reflectivities which cause the concentric rings (line 597)”, the hyporeflective zones can be correspond to the area of cystoid space in the OCTA images. Did you find that? 2) the hyporeflective sings appear irregularly scattered while the reflection of the concentric rings showed regularity. How do you explain the difference?

Although the concentric rings around the macular center appear relatively regular, there is no perfect regularity. The example of Figure 1C (below) shows rings with different distances between individual rings in the lower central macula as well as irregularly arranged parts which seem to link different rings. 

 We found for the left eye of Figure 1B a mean distance of 158.0 ± 8.4 µm between hyperreflective bands in the nasal HFL and a mean distance of 174.2 ± 32.4 µm between individual concentric rings in the nasal macula; both values were not significantly different. The nasal-temporal diameter of the area with concentric rings on OCTA images was 3.3 mm while the nasal-temporal extension of the area with hyporeflective zones on SD-OCT images was 3.6 mm. However, to get comprehensive statistics, we need a larger cohort of subjects with high-grade foveal hypoplasia which we do not have presently. Therefore, we present the possible association between cystoid spaces and concentric rings as assumption and focus our discussion on the possible cause of the cystoid spaces, i.e., thin ONL which results in a thick HFL because the thickness of both ONL and HFL is not altered in eyes with foveal hypoplasia compared to control eyes, as well as possible implications of morphological parameters for the understanding of the development of foveal hypoplasia. 

Additional comments:

1) For statistical analyses (including mean) BCVA should be converted to LogMAR.

We now show BCVA values in logMAR in Table 1 and Figure 3C. 

2) It seems that the OCT image of the 1st low in Fig 2D seems to contain hyporeflective bands (right?).

We mentioned in the Discussion section: "In some eyes with low-grade foveal hypoplasia and control eyes, the HFL may exhibit a striped appearance with alternating vertical medium- and higher reflective bands (e.g., lowest image of Figure 2D)." We discussed this with a possible distortion of the foveal tissue which alters the position of Henle fibers. However, the striped appearance of the HFL on SD-OCT images of some eyes with low-grade foveal hypoplasia and control eyes was caused by alternating medium- and higher reflective bands; hyporeflective bands or zones were not observed on images of these eyes. 

3) I do not understand the merit of the present classification of foveal hypoplasia: a cut off value of 100 um of the thickness of the inner retina. How about using the earlier grading systems based on refs. #26 and #27? According to Fig 2, high and low grades can be grade 1 and 2, respectively.

It is right that the classification of high- and low grade foveal hypoplasia is partly arbitrary. We also write in the manuscript that there is rather a gliding transition of morphological parameters between both groups. However, in addition to the thickness of the inner retinal layers at the macular/foveal center, there are further differences in tissue structure between both hypoplasia groups which we mention in the manuscript. The most important difference was that we found hyporeflective zones in the HFL on OCT images of eyes with high-grade foveal hypoplasia but not on images of eyes with low-grade foveal hypoplasia. Further morphological differences between both hypoplasia groups were, for example, that the thickness of the central ONL was similar in eyes with low-grade foveal hypoplasia and control eyes, but decreased in eyes with high-grade foveal hypoplasia (Fig. 3B), that the thickness of the HFL was significantly different between both hypoplasia groups (Fig. 3B), and that the thicknesses of the central and paracentral ONL were correlated in eyes with high-grade foveal hypoplasia but not in eyes with low-grade foveal hypoplasia (Tab. 3). We mean that our classification system reflects real differences in the morphology of the macular/ center of eyes with high- and low-grade foveal hypoplasia. However, our classification is not perfect. We found, for example, differences in the mean thickness of the HFL among eyes with high-grade foveal hypoplasia (Fig. 5E). In order to expand the statistical analysis in respect to morphological differences among these eyes, we need higher n numbers which we do not have presently. Our results may serve as direction for further investigations of the morphology of the central macula which may help to understand the development of foveal hypoplasia. 

 It is partly difficult to match our classification system with that of Thomas et al. [26]. We mean that "low-grade foveal hypoplasia" matches with grade 1 foveal hypoplasia of Thomas et al. [26]. On the other hand, the eyes with high-grade foveal hypoplasia investigated in our study displayed morphological features which were described by Thomas et al. [26] by grades 2 to 4 foveal hypoplasia. We included the following sentence in the first paragraph of the Results section: "'Low-grade foveal hypoplasia' is comparable with grade 1 foveal hypoplasia of Thomas et al. [26] whereas 'high-grade foveal hypoplasia' includes morphological features which were described by Thomas et al. [26] by grades 2 to 4 foveal hypoplasia." 

4) Although Table 1 shows smaller and larger hyporeflective zones as a category, the exact extent has not been shown. Please add these values in Table 1. All OCT images including Figure 2D should be referrable to Patient IDs. Please add them in the legends (figure reference in Table 1 is unnecessary).

Thank you very much for your suggestion to present the size of the hyporeflective zones in the HFL! However, as we mentioned in the Results section, the shape of the hyporeflective zones on SD-OCT images varied which made it difficult to obtain precise values of the size. On the other hand, because many hyporeflective zones extended through the entire HFL, the thickness of the HFL should be different between eyes with high-grade foveal hypoplasia and larger and smaller hyporeflective zones in the HFL. Indeed, the central and paracentral HFL were significantly thicker in eyes with larger hyporeflective zones compared to eyes with smaller zones (new Figure 5E). We also show individual values in Figure 5E; therefore, we want to avoid a mention of these values in Table 1. We write also in the Results section that "the central HFL of eyes with larger hyporeflective zones had a thickness of 49‒70 µm whereas that of eyes with smaller zones was 20‒46 µm". 

 We deleted the figure reference from Table 1 and mention the subject IDs in the legends of Figures 1 and 2. 

5) Determining of the width of HFL at the foveal center is challenging in eyes with foveal pits because of the direction of the incident light: the thickness of HFL in eyes with low grade foveal hypoplasia and control seems inaccurate. The authors need to provide how the measurement was obtained accurately. Line 210: “the thickness of individual retinal layers was manually measured using the Heidelberg” is not enough (where is the border between HFL and ONL in Figure 3A right?). 

We wrote in the first version of the manuscript that one selection criterium for control eyes was the detectability of the central and nasal ONL on the SD-OCT images. We added the half-sentence that this means that the HFL displayed a higher reflectivity than the ONL. Of course, in most SD-OCT images, HFL and ONL cannot be differentiated because both layers display the same reflectivity. However, there are also single SD-OCT images which show a higher reflectivity of the HFL compared to the ONL, likely because the direction of the incident light favored such reflectivity difference. We searched for SD-OCT images of control eyes (without indications of retinal and choroidal disease) which showed a higher reflectivity of the central and nasal HFL; we used SD-OCT images of 75 of such control eyes to measure the thicknesses of the ONL and HFL at the center and 0.5 and 1.0 nasal from the center. 

 We changed some images of Figure 2D to show more scans in which the nasal HFL displays a higher reflectivity than the ONL. 

 We altered Figure 3A because we remeasured the photoreceptor length; the distances which we measured for the new version of the manuscript are shown in Figure 3A and explained in the M&M section. In Figure 3A right, there is a thin layer of hyporeflectivity between the hyperreflective inner layer and the medium-reflective ONL. The hyperreflective inner layer represents the horizontal layer of the Müller cell cone. According to histological studies, the thin layer of hyporeflectivity represents the thin HFL in the center of the fovea.

---

## [Decision Letter · Decision Letter 1]

31 Mar 2022

Morphology of foveal hypoplasia: Hyporeflective zones in the Henle fiber layer of eyes with high-grade foveal hypoplasia

PONE-D-21-37391R1

Dear Dr. Bringmann,

We’re pleased to inform you that your manuscript has been judged scientifically suitable for publication and will be formally accepted for publication once it meets all outstanding technical requirements.

Kind regards,

Tudor C. Badea, M.D., M.A., Ph.D.

Academic Editor

PLOS ONE

Additional Editor Comments (optional):

Reviewers' comments:

Reviewer's Responses to Questions

**Comments to the Author**

1. If the authors have adequately addressed your comments raised in a previous round of review and you feel that this manuscript is now acceptable for publication, you may indicate that here to bypass the “Comments to the Author” section, enter your conflict of interest statement in the “Confidential to Editor” section, and submit your "Accept" recommendation.

Reviewer #1: All comments have been addressed

2. Is the manuscript technically sound, and do the data support the conclusions?

Reviewer #1: Yes

3. Has the statistical analysis been performed appropriately and rigorously? 

Reviewer #1: Yes

4. Have the authors made all data underlying the findings in their manuscript fully available?

Reviewer #1: Yes

5. Is the manuscript presented in an intelligible fashion and written in standard English?

Reviewer #1: Yes

6. Review Comments to the Author

Reviewer #1: The authors have logically responded according to the review comments. For the descriptive statistics, the average of the BCVA should not be calculated by raw values (angles) but LogMAR be used: this has been corrected in the revised version.

7. PLOS authors have the option to publish the peer review history of their article (what does this mean?). If published, this will include your full peer review and any attached files.

Reviewer #1: No

---

## [Editor Report · Acceptance letter]

4 Apr 2022

PONE-D-21-37391R1 

Morphology of foveal hypoplasia: Hyporeflective zones in the Henle fiber layer of eyes with high-grade foveal hypoplasia  

Dear Dr. Bringmann:

I'm pleased to inform you that your manuscript has been deemed suitable for publication in PLOS ONE. Congratulations! Your manuscript is now with our production department. 

Kind regards, 

on behalf of

Dr. Tudor C. Badea 

Academic Editor

PLOS ONE